# Data-driven grading of acute graft-versus-host disease

Evren Bayraktar [1,2,3,13], Theresa Graf[1,2,13], Francis A. Ayuk [4], Gernot Beutel [5], Olaf Penack [6], Thomas Luft [7], Nicole Brueder [5], Gastone Castellani [8], H. Christian Reinhardt[2,9,10], Nicolaus Kröger [4], Dietrich W. Beelen [2,9] & Amin T. Turki [1,2,9,10,11,12] ✉

Despite advances in allogeneic hematopoietic cell transplantation, acute graft-versus-host disease (aGVHD) remains its leading complication, yet with heterogeneous outcomes. Here, we analyzed aGVHD phenotypes and clinical classifications in depth in large, multicenter cohorts involving 3019 patients and addressed prevailing gaps by developing data-driven models. We compared, tested and verified these along with all conventional classifications in independent cohorts and found that data-driven grading outperformed conventional grading in Akaike information criterion and concordance index metrics. Data-driven classifications refined aGVHD assessment with up to 12 severity grades, which were associated with distinct nonrelapse mortality (NRM) and confirmed the key role of intestinal aGVHD. We developed an online calculator for physicians to implement principal component-derived grading (PC1). These results provide substantial insight into the evaluation of aGVHD phenotypes and multiorgan involvement, which relegates the exclusive reporting of overall aGVHD severity grades in transplant registries and clinical trials. Data-driven aGVHD grading provides an expandable platform to refine classification and transplant risk assessment.

Acute graft-versus-host disease (aGVHD) remains the major cause of early morbidity and nonrelapse mortality (NRM) after allogeneic hematopoietic stem cell transplantation (HCT)[1]. It's clinical grading, leveraging the physician's assessment of 3 target organs (skin, liver, intestine), was introduced by Glucksberg et al.[2], later revised by the Keystone GVHD Consensus conference[3], by the International Blood and Marrow Transplant Registry (IBMTR)[4,5], the University of Minnesota[6,7] and by the Mount Sinai aGVHD International Consortium (MAGIC)[8].

[1]Computational Hematology Lab, West-German Cancer Center, University Hospital Essen, Hufelandstr. 55, 45122 Essen, Germany. [2]Department of Hematology and Stem Cell Transplantation, West-German Cancer Center, University Hospital Essen, Hufelandstr. 55, 45122 Essen, Germany. [3]Chair III of Applied Mathematics, TU Dortmund University of Applied Sciences, Vogelpothsweg 87, 44227 Dortmund, Germany. [4]Department for Stem Cell Transplantation, University Medical Center Hamburg-Eppendorf (UKE), Martinistraße 52, 20251 Hamburg, Germany. [5]Department of Hematology, Hemostasis, Oncology and Stem Cell Transplantation, Hannover Medical School, Carl-Neuberg-Str. 1, 30625 Hannover, Germany. [6]Department of Hematology, Oncology and Tumorimmunology, Charité – Universitätsmedizin Berlin, corporate member of Freie Universität Berlin and Humboldt-Universität zu Berlin, Augustenburger Platz 1, 13353 Berlin, Germany. [7]Department of Internal Medicine V, University Hospital Heidelberg, Im Neuenheimer Feld 410, 69120 Heidelberg, Germany. [8]Department of Medical and Surgical Sciences- DIMEC, Applied Physics and Biophysics group, University of Bologna, Via Zamboni 33, 40126 Bologna, Italy. [9]German Cancer Consortium (DKTK), Partner sites Essen/Düsseldorf, Hufelandstr. 55, 45122 Essen, Germany. [10]Cancer Research Center Cologne Essen (CCCE), Partner site Essen, Hufelandstr. 55, 45122 Essen, Germany. [11]Department of Hematology and Oncology, Marienhospital University Hospital, Ruhr-University Bochum, Universitätsstr. 150, 44801 Bochum, Germany. [12]Institute for Experimental Cellular Therapy, University Hospital Essen, Hufelandstr. 55, 45122 Essen, Germany. [13]These authors contributed equally: Evren Bayraktar, Theresa Graf. ✉e-mail: amin.turki@uk-essen.de

Today, in particular, multiorgan involvement and therapy resistance remain a challenge in aGVHD assessment. Inconsistent assessment practices between HCT centers and unwitting use of different grading systems further reduce the comparability of data[9,10]. During the last decade, many efforts to improve aGVHD assessment and outcome prediction have focused on the identification of universal[11,12] and organ-specific aGVHD biomarkers[13–15]. Indeed, biomarker combinations can predict 6-month nonrelapse mortality (NRM), aGVHD mortality[16,17], and overall survival (OS) in aGVHD patients[18] but are not universally available. Accumulating evidence of complex aGVHD biology[19] conflicts with the quest for a simple assessment and classification for clinical practice.

More recently, promising efforts have been undertaken employing machine learning methods to predict mortality[20,21] and aGVHD[22–24] after HCT. Given these recent insights, prevailing discrepancies in conventional aGVHD classification practice, and substantially varying survival outcomes of patients with the same aGVHD severity grade, we hypothesized that a data-driven approach could shed light on the strengths and limitations of aGVHD classifications and respond to ongoing issues such as multiorgan involvement, heterogeneous phenotypes and their relation to outcomes. We leveraged several data science methods (including derivatives of principal component analysis (PCA), hierarchical clustering (Hclust), *K*-means clustering, uniform manifold approximation and projection (UMAP) and density-based spatial clustering of applications with noise (DBSCAN) on clinical aGVHD organ assessments to address these issues in a large multicenter cohort of HCT patients with aGVHD. These grading methods resulting from our work were independently validated and tested (an overview is provided in Fig. 1).

## Results

Starting from the hypothesis that an in-depth data-driven analysis of clinical aGVHD phenotypes (i.e., organ severity combinations/involvements) and grading practices may improve the understanding of aGVHD and support optimal treatment strategies, we standardized reporting between centers via full documentation of aGVHD organ involvement while assembling a multicenter aGVHD dataset and analyzing it to an unprecedented extent. Using unsupervised learning methods, we then developed several data-driven aGVHD grading systems and compared their severity indexing along with that of conventional aGVHD grading systems via different performance indices.

This large, multicenter dataset included contemporary HCT patients diagnosed with aGVHD from 5 major German HCT centers ($n = 3019$). Aiming for a proportional split of 2/3 and 1/3 between the training set and the independent test set, aGVHD patients from two centers were used for training (Essen, Hamburg; $n = 2319$) and three independent centers for testing (Berlin, Hannover, Heidelberg; $n = 700$). The cohorts' baseline characteristics were balanced for age, sex, recipient cytomegalovirus-positive serostatus (CMV R+), donor type, and TBI and are detailed in Supplementary Table 1. We noted differences in the proportion of some diseases and myeloablative conditioning. Exploratory data analysis confirmed a globally conserved data structure and equal proportions of aGVHD phenotypes in both the training and test sets (Fig. 2a, c), supporting an appropriate split despite expected minor differences. Both the pair plots from kernel density estimation and the Spearman correlation had comparable shapes to those of a random 2/3 and 1/3 dataset splitting (Supplementary Fig. 1a, b and c, d), while preserving the independent character of these sets via distinct HCT sites. In both cohorts, the Spearman correlation matrix revealed a concordance of combined aGVHD liver and GI involvement of ~20%, while skin and GI involvement were negatively associated (Fig. 2b, d).

### Development of a data-driven grading system

For the development of data-driven aGVHD grading systems using unsupervised learning, we gathered data on individual patients' organ involvement in a multidimensional data space (one dimension for each aGVHD target organ: skin, liver and gastrointestinal tract (GI), with organ-specific aGVHD severity represented in the range from 0 to 4) with the intention to represent the maximum of the clinical variance of aGVHD (i.e., the data spread of the 125 (=$5^3$) possible combinations, with zero corresponding to the absence of aGVHD in the given organ). Due to a linear variation in the data (Fig. 3a), we started with a linear principal component analysis (PCA) to derive a data-driven grading. We aimed to map the multidimensional data to a one-dimensional space to formulate an aGVHD severity index (Eq. (1)). Hence, we applied PCA dimensionality reduction to the dataset, with the first of the three principal components (PC1) explaining approximately half of the overall variance (Fig. 3b). We plotted all individual patients in the training set ($n = 2319$) according to PC1 and tested their separation on the PC1 axis by coloring for MAGIC (Fig. 3c) or Consensus grades (Supplementary Fig. 2a). Indeed, conventional severity grades separated data on the PC1 axis despite some overlap. Next, the continuous PC1 axis was transformed into 12 PC1-derived severity stages (Eq. 2). When graphically plotted against OS, increasing PC1 stages were correlated with decreasing long-term OS (Fig. 3D), which we also approximated by a power function (Supplementary Fig. 2b). When condensed into four grades as in conventional aGVHD grading systems, PC1-derived grading separated four distinct strata for 12-month OS ($p < 0.0001$, Fig. 3e). We performed several tests to validate each of the data-driven aGVHD grading algorithms. First, we ran multiple benchmarks on our training set, including 500-fold resampling, which provided stable results with small confidence intervals (Supplementary Fig. 1c). When we plotted PC1 stages against aGVHD organ involvement categories, we observed higher cumulative numbers of aGVHD organ stage involvement (i.e., multiorgan involvement) in advanced PC1 severity stages (Fig. 3f, Supplementary Fig. 1d). Isolated skin aGVHD was exclusive to the lowest PC1 stages, followed by liver or GI involvement. The most common types of organ involvement were isolated skin, isolated GI and the combinations of skin and GI aGVHD. As an alternative approach to PC1 grading, we leveraged unsupervised clustering methods (hierarchical clustering (Hclust) and *K*-means). Indeed, Hclust successfully dissected the training data into four clusters with significantly distinct OS (Fig. 4a, b). *k*-means clustering also successfully distinguished aGVHD phenotypes in the training cohort, and its performance was measured by the sum of squares of distances (SSD) and silhouette coefficient (Fig. 4c, d). Again, for both clustering methods, we set the cutoff at four clusters to have a format comparable to conventional grading. However, both the SSD elbow method and the best metrics ratio (silhouette index of 0.62/8 clusters) determined an optimal number of *k*-means clusters of 8, indicating that using >4 severity grades may provide additional value. In addition to these linear methods, we explored common nonlinear approaches (UMAP and t-SNE) on the data to address heterogeneity, and the results are detailed in the Supplementary Notes and Supplementary Figs. 3–5.

### Verification and test on independent data

For verification of the data-driven grading on the independent test data, we employed several quality indices, analyzed the proportional patient distribution and the association of aGVHD severity grades with clinical outcomes to describe the exactness and spread of differences between classifiers. First, we built multivariate competing risk and Cox regression models for NRM (Fig. 4e) and OS (Supplementary Fig. 6) considering the aGVHD severity assessed by the PC1-based grading as a time-dependent covariate, which remained significant for independent test data and after including potential confounders as covariates. Significant covariates in the adjusted NRM model were the diagnoses of acute lymphoblastic leukemia (ALL), myelodysplastic syndromes (MDS), other diagnoses, the year of HCT and the EBMT risk score. In the next step, we comparatively examined the other data-driven

## a) Data preparation

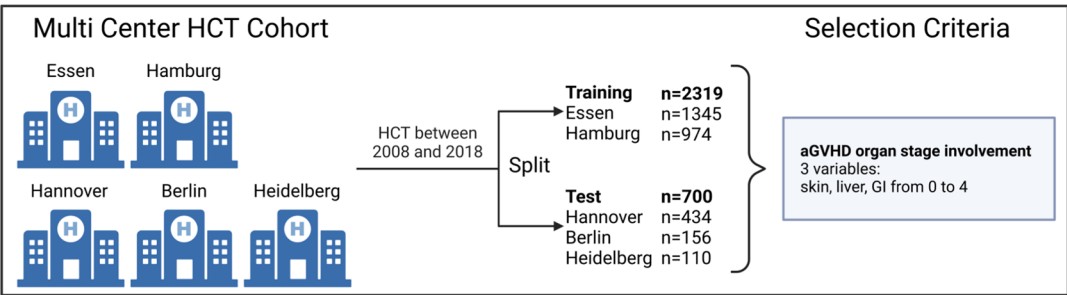

## b) Data-driven aGVHD classification

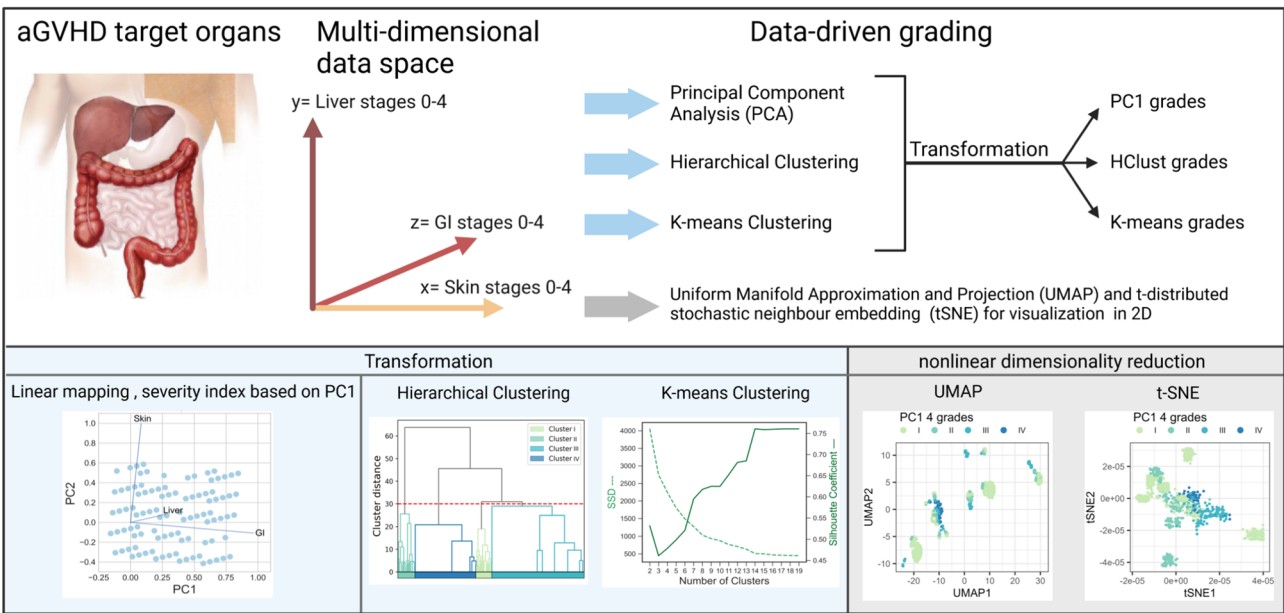

## c) Evaluation, validation, test and verification of data-driven aGVHD grading

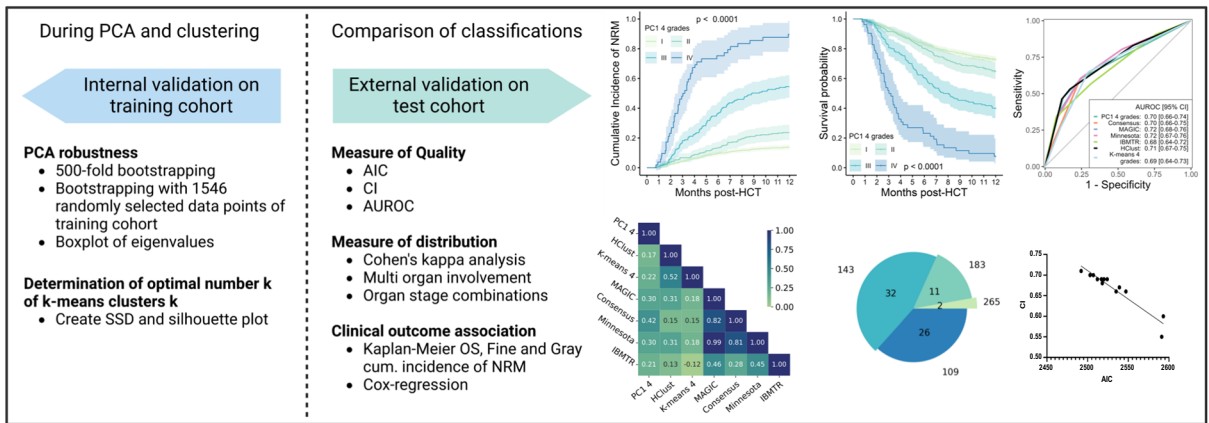

(Fig. 5a–c) and conventional (Fig. 5d–f) aGVHD grading systems, which each stratified the independent test cohort (*n* = 700) into groups with significantly distinct OS values (*p* < 0.0001). Starting from the day of aGVHD diagnosis, the NRM curves also revealed significant differences (*p* < 0.0001, Supplementary Fig. 7). Using an HCT cohort without aGVHD as a common Cox reference illustrated the distribution of hazard ratios in each examined data-driven and conventional classification (Supplementary Fig. 8). The time-dependent AUROC curves for 12-month OS and NRM varied from 0.68 (95% confidence interval (CI) 0.64–0.72) with IBMTR to 0.72 (95% CI 0.68–0.76) with MAGIC and from 0.72 (95% CI 0.67–0.77) to 0.77 (95% CI 0.72–0.81), respectively.

The AUROC of the PC1 grading was comparable to that of the Consensus and numerically higher for NRM (Fig. 5g, h). Despite careful consideration of the time-dependent character of aGVHD in these models, the clinical outcome association alone may not solve the issue of classification. We next leveraged Cohen's kappa analysis to reveal the intergrading agreement between classification systems, which was strongest between MAGIC and Minnesota and least present between HClust and IBMTR grading (Fig. 5l). Except for IBMTR, the intergrading agreement was very high among conventional grading systems. Data-driven grading systems, however, were distinct from conventional systems and differed from each other. Given the differences in Cohen's

**Fig. 1 | Overview of data-driven aGVHD grading development, validation, external test and verification in comparison to conventional grading. a** Data preparation: Data was assembled from a multicenter cohort (Berlin, Essen, Hamburg, Heidelberg, Hannover) with HCT between 2008 and 2018 and split into independent training ($n = 2319$) and test cohorts ($n = 700$). **b** Data-driven aGVHD classification. Input data from the aGVHD target organ involvement (skin, GI and liver) was organized in a 3D space and the following data-driven methods were applied: Principal component analysis (PCA) for linear mapping of PC1 and severity indexing, as well as hierarchical- and $k$-means clustering. For comparability with conventional grading, the number of clusters was set to 4. The nonlinear methods Uniform Manifold Approximation and Projection (UMAP) and $t$-distributed stochastic neighbor embedding (tSNE) were used to visualize grading in 2-dimensional space. Non-linear methods and their results are detailed in the supplement. **c** Evaluation, validation, external test, and verification of data-driven aGVHD grading. PCA was internally validated via 500-fold bootstrapping of 1546 randomly selected data points (2/3 of training cohort). During $k$-means clustering, the optimal cluster number was determined using the elbow method on the sum of squared distances (SSD) and silhouette index. All grading systems were externally tested on independent multicenter data. Akaike information criterion (AIC) as well as the concordance index (Ci) were calculated to verify and compare data-driven and conventional grading. Time-dependent AUROC curves (Area under the receiver operating characteristic curves) were generated to visualize specificity and sensitivity for 12-month OS and NRM. Distribution plots and Cohen's kappa analysis compared the distribution of the different phenotypes (organ-stage combinations) and intergrading agreement. Kaplan–Meier OS and cumulative incidence NRM curves were computed with 95% confidence intervals (CI) to compare associations of different grading systems with outcome. *P*-values were calculated using a two-sided log-rank test (Kaplan–Meier OS) or two-sided Gray test (NRM curves). Created with BioRender.com. The organs image in panel **b** is adapted from https://pixabay.com/illustrations/offal-marking-medical-colon-liver-1463369/ via Elionas2 under the Content License.

kappa, we compared the proportion of distinct aGVHD phenotypes (=organ stage involvement) in each system's grading, beyond their comparable capacity to significantly separate OS or NRM strata (Fig. 6). To further analyze the repartition (size and proportions) of these aGVHD phenotypes within each severity grade, we comparatively plotted MAGIC and data-driven grading systems along with their NRM curves to explore the spread of differences between classifiers. The MAGIC NRM curves distinguished significantly between grades III and IV but not between grades I and II (Fig. 6a). As indicated by Cohen's kappa, the phenotype distribution differed between classifications. MAGIC grading (Fig. 6a–c) included two phenotypes for grade I with 265 patients (Fig. 6b), which were distributed 50%-50% (Fig. 6c), whereas it involved 26 phenotypes for 109 patients for grade IV. PC1 grading (Fig. 6d–f) classified 17 phenotypes with 443 patients into grade I (Fig. 6e), including four large phenotypes each representing >10% of the grade I patients (Fig. 6f). PC1 significantly separated the NRM strata between grades I and II but also between grades III and IV (Fig. 6d); its phenotypes were almost equally distributed across aGVHD grades. Hierarchical clustering (Fig. 6g–i) had fewer organ combinations in its lower severity grades I and II than $K$-means (Fig. 6j–l), but neither distinguished NRM between grades I and II (Fig. 6g, j). With a total of 32 phenotypes, MAGIC grade III assembled the largest number of aGVHD phenotypes in a single grade, followed by Hclust (23 phenotypes) and PC1 (22 phenotypes). The complete data, including phenotypes and their proportions within each aGVHD grade, are provided in Supplementary Data 1–7. The leading phenotypes among MAGIC grade III patients were isolated GI stages 2 or 3 (Supplementary Data 1). Within the grade IV phenotypes, the organ combinations of GI stages 3–4 without other organ involvement (Hclust, Supplementary Data 3) and combinations of GI and skin aGVHD (e.g., skin stage 3 + GI stage 4, $K$-means, Supplementary Data 4) assembled most patients. Altogether, this repertoire analysis provided detailed insights into each classification's preferences for phenotype sets but also revealed inconsistencies, e.g., of the clustering-based aGVHD grading systems (Hclust and $K$-means), which graded, e.g., combinations of GI and other organs at a lower severity than single GI stage 1 involvement (Supplementary Data 4). This limitation, however, was not observed for the PC1 grading system.

Given that MAGIC grade III assembled the highest number of aGVHD phenotypes (Fig. 6a, b), we suspected heterogeneity within this grade and wondered if the data-driven grading could identify distinct patient subgroups within MAGIC grade III in the test cohort. For this purpose, we segregated MAGIC grade III patients for whom PC1 calculated less severe aGVHD grades from the remaining grade III MAGIC patients (Fig. 7a) and found that the redistributed patients actually had a significantly higher OS ($p = 0.0049$) than the remaining grade III MAGIC patients. Similar heterogeneity was observed for Consensus grade III patients (Fig. 7b). Differences in OS for redistributed patients remained significant when further stratified into PC1 grade I and PC1 grade II (Fig. 7c, d). Analysis of 12-month NRM confirmed significant differences for redistributed grade III Consensus patients, not for MAGIC, which only differed significantly for 6-month NRM. Importantly, these redistributed grade III MAGIC patients had specific phenotypes with respect to GI or multiorgan involvement. Patients lacking stage 3 GI involvement or stage 3 liver involvement in combination with GI stage 2 or the combination of stage 3 skin with stage 2 liver and stage 2 GI were categorized as PC1 grade II (Supplementary Data 6). Taken together, these results showed that data-driven algorithms discriminate aGVHD phenotypes differently, which may be beneficial for further dissecting the heterogeneity of phenotypes within conventional grades. To provide some insights into underlying biological differences for the classification process, we stratified the test cohort by one organ system severity (either skin, liver or GI, also in patients with multiorgan involvement). For example, patients with stage 1 skin, stage 2 liver and stage 2 GI involvement would be represented as skin 1 (Fig. 8a, b), liver 2 (Fig. 8c, d) and GI 2 (Fig. 8e, f). Here, the severity of GI involvement was the best single organ indicator to significantly discriminate OS or NRM in the test cohort (Fig. 8e, f). The low OS of patients with stage 0 skin involvement was unexpected (Fig. 8a) but explained by them having multiorgan aGVHD involving the GI tract or liver. Despite its overall significance for OS ($p < 0.0001$), skin aGVHD alone was the least effective in distinguishing patient cohorts with respect to outcome. Accordingly, the PC1 loadings plots of the training cohort (Fig. 3a) credited only 8% of its weight to skin and 96% to GI involvement, explaining the solid performance of the PC1 grading method, which attributes higher importance to GI severity while preserving the importance of multiorgan involvement in its classification.

## aGVHD classifications beyond four grades

Following the hypothesis that higher levels of detail in data-driven grading might improve the performance of classifications, we evaluated several approaches employing more than 4 severity grades. First, we employed all 12 PC1 stages to categorize aGVHD patients, which were associated with a highly diverse separation of NRM and fewer phenotypes per category (Fig. 9a–c). Given that 12 groups became difficult to conceive for human operators, we next tried to simplify this approach by skipping a multiplication operation in Eq. 2, which resulted in a PC1 grading system of six grades (Fig. 9d–f). Both separated the test cohort into proportionally distributed categories with significant differences in NRM. As the intersection of the SSD elbow and silhouette index supported the use of eight $K$-means clusters in the development cohort (Fig. 4c), we also tested this system on the independent test cohort and found that the 8 clusters also separated distinct strata (Fig. 9g–i). The repartition of aGVHD phenotypes in these refined models is detailed in Supplementary Data 9. These results reveal that the great heterogeneity of aGVHD phenotypes, which are associated with distinct clinical

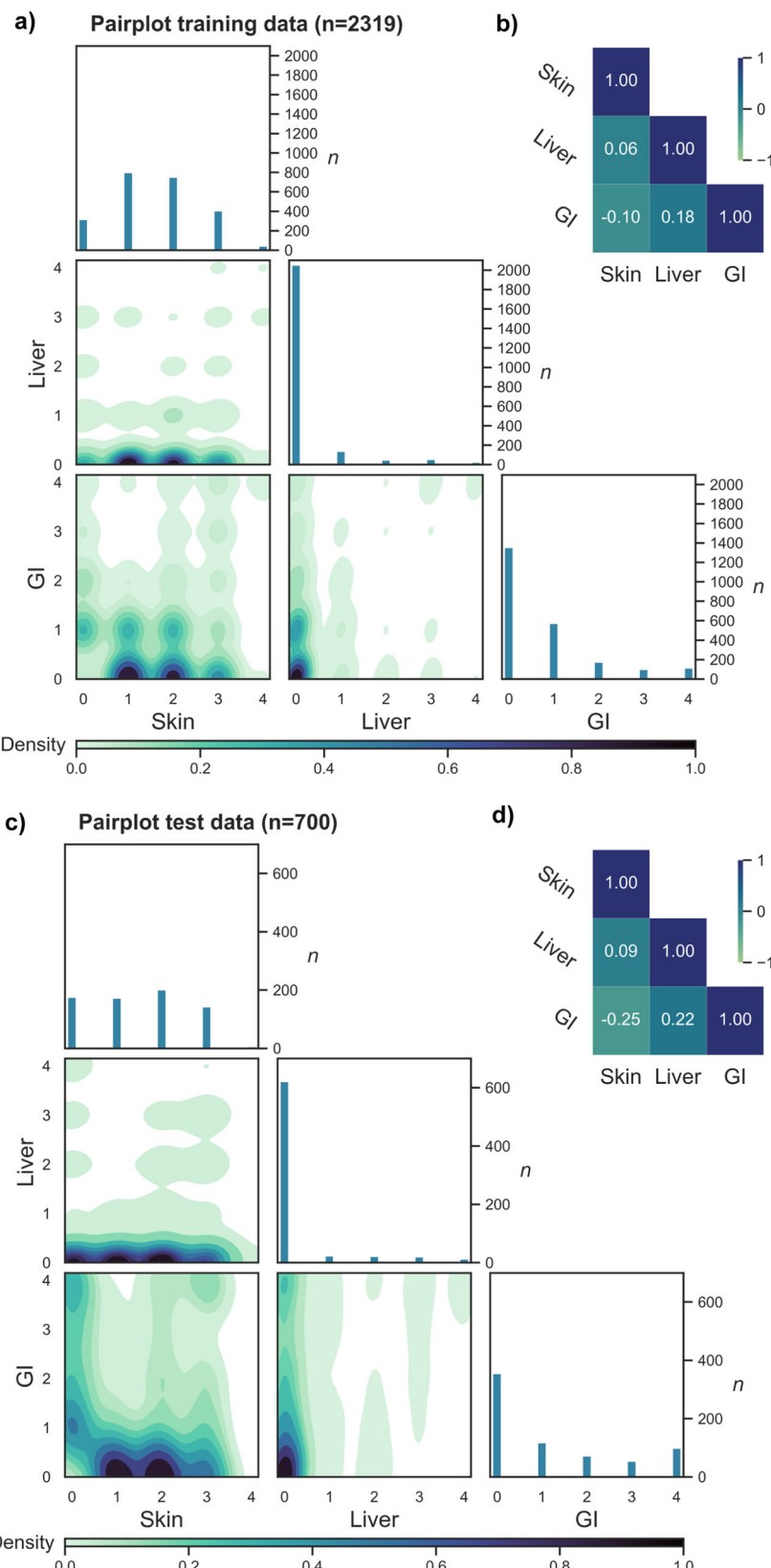

**Fig. 2 | Exploratory data analysis of aGVHD training (*n* = 2319) and test cohorts (*n* = 700) shows adequate cohort coverage. a** Pair plot from kernel density estimation of the training cohort (*n* = 2319) plotting the clinical target organ stages of skin, liver and GI involvement (stages 0-4, left to box and below). The target organ stage correlations are presented as density plots. Patient numbers (*n*) of each subgroup are indicated right in each box. A higher *n* in each subgroup is shown by greater surface coverage. Density of aGVHD target organ combinations is indicated from light green to dark blue. **b** Target organ stage correlation matrix (Spearman) of the training cohort shows the distribution of single variables skin, liver and GI and their respective interactions. Range from −1.0 to +1.0, dark blue indicates full overlap. **c** Pair plot and **d** Target organ stage correlation matrix (Spearman) of the test cohort (*n* = 700). Analysis, labels and colors as in (**a** and **b**).

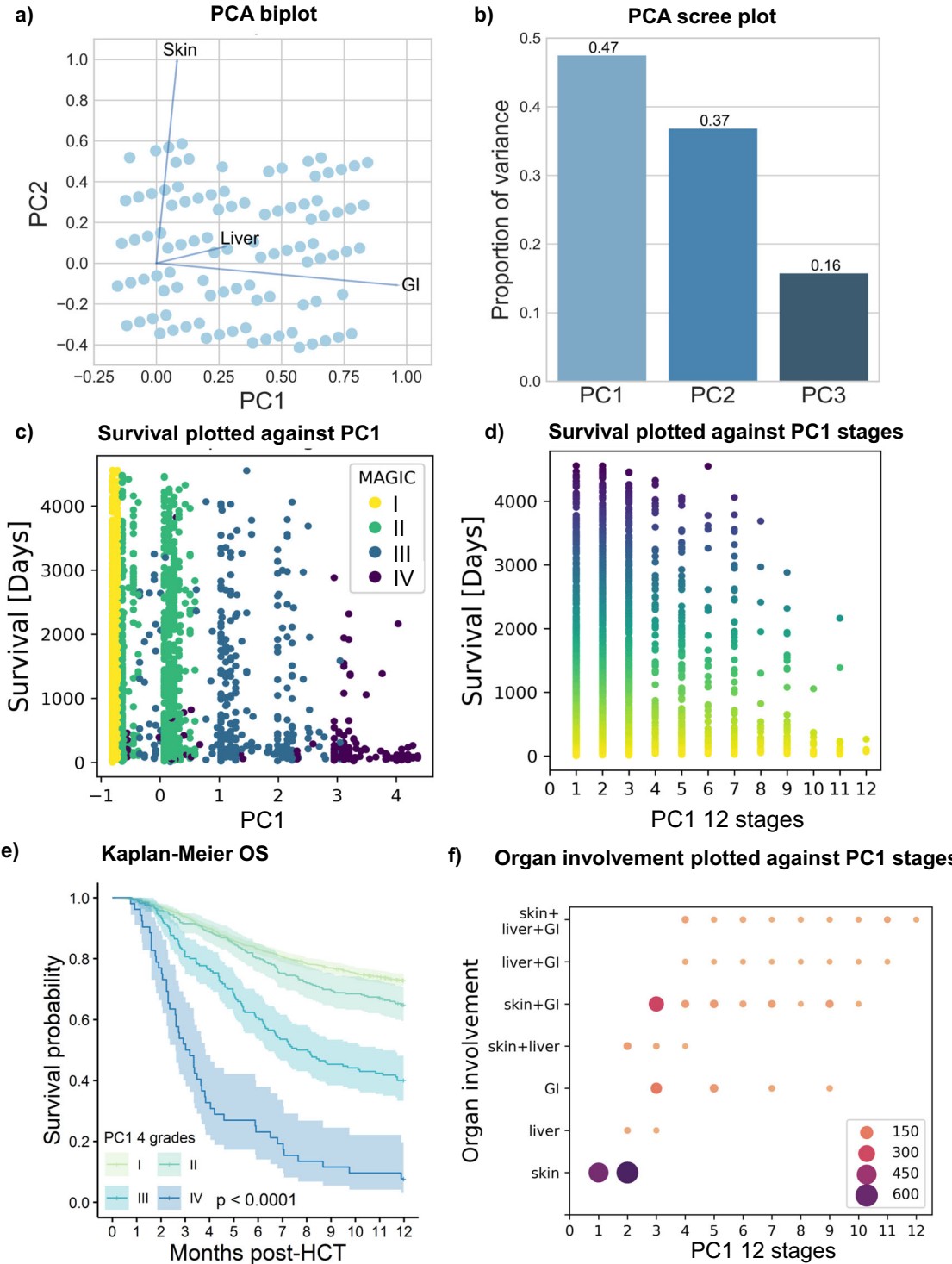

**Fig. 3 | Principal component analysis of the training cohort ($n$ = 2319) and transformation of principal component 1 into PC-grading of aGVHD. a** Biplot of principal components 1 (PC1) and 2 (PC2) on each axis displays the scores and loading vectors of principal component analysis (PCA). Arrows indicate the importance of each target organ involvement for PC1 and PC2, respectively. **b** Scree plot of PC1, PC2 and PC3. The proportion of variance explained by PC1 is the greatest with 0.47. **c** Explorative plotting of PC1 against overall survival (OS, days from HCT, censoring has not been considered in this representation) indicates lower long-term OS with increasing PC1. Each dot represents one patient with aGVHD. Colors representing MAGIC aGVHD grade I–IV (I = yellow, II = green, III =

blue, IV = violet) indicate the overlap of different MAGIC grades. **d** Transformation of PC1 results into an aGVHD classification (ranging from PC1-stage 1–12), results plotted against OS, as in (**c**). Lighter colors (yellow) indicate shorter observation, darker colors (blue) higher long-term OS. **e** Kaplan-Meier estimate OS curve with 95% CI of 4 PC-aGVHD grades (I–IV) consolidated from PC-aGVHD-stages 1–12. The colors indicate lower (yellow) to higher (blue) OS. Strata are compared with the two-sided log-rank test. **f** Plotting of PC1 stages against aGVHD organ involvement (combinations: Skin: only skin; liver: only liver; GI: only GI; skin and liver; skin and GI; liver and GI; skin, liver and GI). The circle size corresponds to the $n$ of patients in each category.

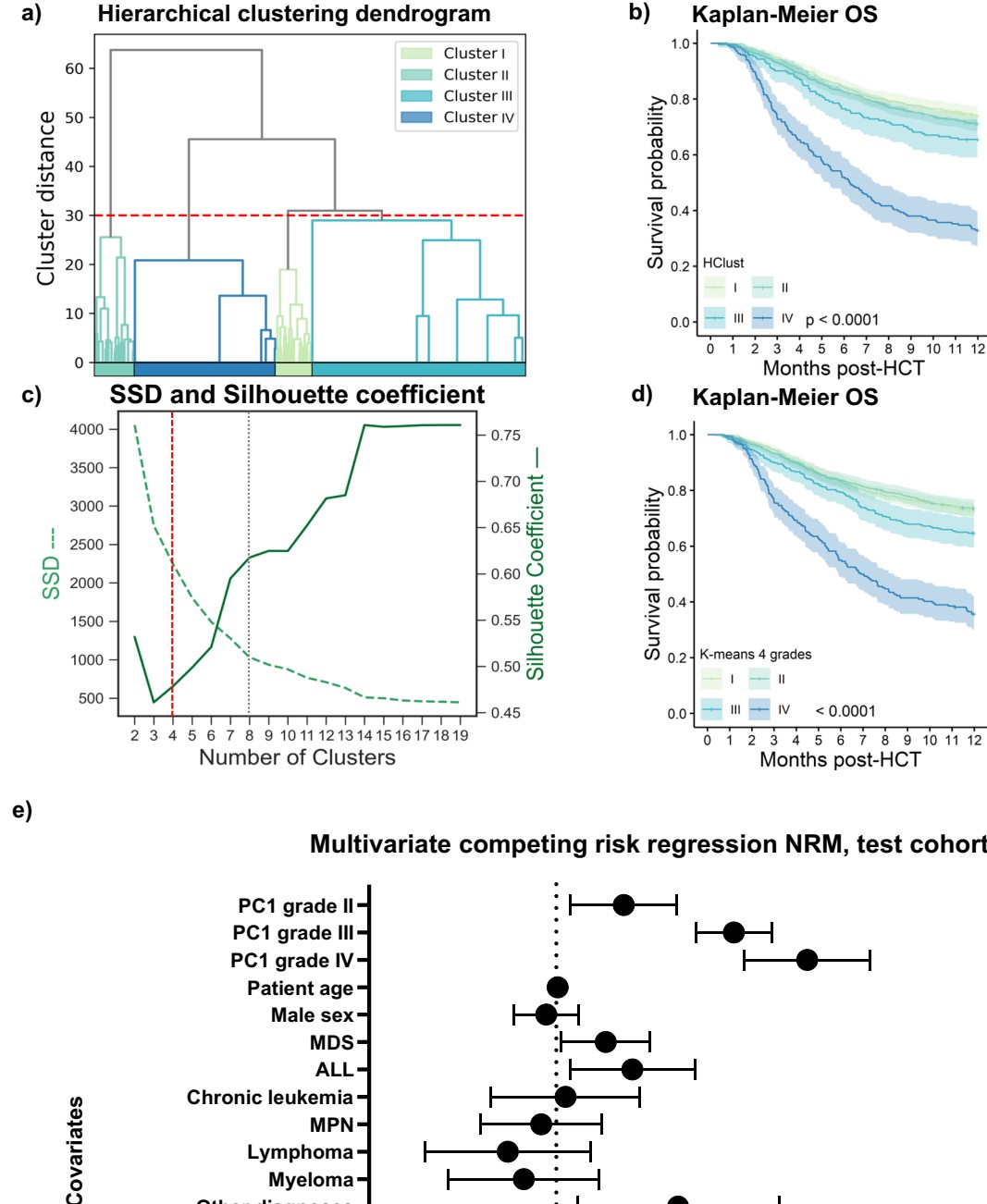

a) **Hierarchical clustering dendrogram**

b) **Kaplan-Meier OS**

c) **SSD and Silhouette coefficient**

d) **Kaplan-Meier OS**

e) **Multivariate competing risk regression NRM, test cohort**

outcomes, may be captured by more granular grading systems covering more than the usual 4 grades.

As the last verification step in the external testing process of data-driven grading, we compared and ranked all grading systems by their performance metrics using the Akaike information criterion (AIC) and concordance index (Ci). The lowest−and therefore best−AIC was

found among the data-driven grading systems (Fig. 10a). Among the conventional grading systems, MAGIC had the best AIC. There was high agreement between the AIC and Ci in prioritizing data-driven classifications, particularly PC1 grading (Fig. 10b, c). Taken together, these results indicate that several criteria, including the phenotype composition, performance metrics and outcome association, should

**Fig. 4 | Hierarchical and partitional clustering of the training cohort (n = 2319) as alternative data-driven approaches to aGVHD grading and multivariate competing-risk-regression of the validation cohort (n = 700) with the PC1 grading. a** Agglomerative hierarchical clustering (HClust) dendrogram of the training cohort on the basis of their target organ involvements. An HClust distance threshold of 30 split the cohort into 4 clusters, which were numerically ordered. Grade I: n = 763; II: n = 1149, III: n = 191, IV: n = 216. Red dashed line indicates cutoff level for four grades. **b** Kaplan–Meier OS curve with 95% CI of 4 HClust-aGVHD grades (I–IV). Strata are compared with the two-sided log-rank test. **c** K-means partitional clustering performance indicators SSD (sum of squared distances, green dashed) and silhouette coefficient (green), labels on each figure side. Red dashed line indicates cutoff level with four grades; gray dashed line shows cutoff level with 8 grades, the optimal number determined by both methods (n = 8, Sil = 0.62). We evaluated a further cutoff point with 14 clusters in the supplementary notes. **d** Kaplan–Meier OS curve with 95% CI of K-means-4 grades (I–IV). Strata are compared with the two-sided log-rank test. **e** Multivariate competing risk regression

analysis for 12 months NRM on the test cohort (n = 541 evaluable for all covariates) using the PC1 aGVHD grades as a time-dependent variable. The multivariate model was adjusted for potentially confounding variables, covariates as listed in **e**. Horizontal bars represent 95% CI. P-values are computed based on the Wald-test. The hazard ratio (HR) is a measure of the ratio of the hazard between two groups. A value of 1 is the reference, HR < 1 corresponds to lower risk and HR > 1 to higher risk of NRM than the reference. The HR of PC1 grade II was 2.12 (95% confidence interval, CI, 1.17-3.83, grade III HR 7.2 (95%CI 4.72-10.99) and grade IV HR 16.30 (95%CI 8.12-32.75). Significant covariates in this NRM model were diagnoses (acute lympho-blastic leukemia (ALL) HR 2.3 (95%CI 1.17–4.66), myelodysplastic syndromes (MDS) HR 1.74 (95%CI 1.06–2.84), other diagnoses HR 3.8 (95% CI 1.27–11.88), year of HCT HR 0.91, 95% CI 0.85–0.97, and EBMT risk score HR 1.40, 95%CI 1.21–1.63. The covariates, donor age, donor sex, donor type, Karnofsky performance index ≥80 were not significant in univariate regression analysis and hence not included in the multivariate model. Source data are provided as a source data file.

be taken into consideration to assess grading quality and to better understand the strengths and weaknesses of each approach. They also confirm that MAGIC is a good grading system. Most importantly, the data-driven grading approaches—in particular, PC1 refined to 6 or 12 grades—provide additional value beyond current practice in differentiating aGVHD phenotypes and globally lead the classification quality indices.

## Discussion

This study is the first to systematically evaluate conventional aGVHD grading along with data-driven methods and presents results from the development, validation, and independent test of the first data-driven aGVHD grading systems. Via extensive comparisons of the different strategies, this study answers the question of which domains data-driven classifications reveal advantages in over conventional classifications and vice versa. Our results shed light on the complexity of clinical aGVHD phenotypes, dissect the impact of multiorgan involvement and may improve the stratification strategies of future clinical trials to assign novel drugs to best-fitting recipients.

Despite new drugs and preventive strategies[25], multiorgan involvement and treatment resistance remain key challenges to the current aGVHD treatment setting, and heterogeneity in reporting is one of the issues. Past efforts to standardize aGVHD grading practices[26] and achieve comparability between HCT centers have positively impacted quality assurance in the HCT community. Our data demonstrate that data-driven grading of aGVHD is feasible, can be implemented via a digital application and reveals several advantages over conventional grading. Data-driven grading also facilitates physicians in managing the complexity of aGVHD phenotypes by weighting and combining multiorgan involvement and finally reducing it to a single score. Overall, data-driven grading outperformed conventional classifications in performance indices. However, it is insufficient to evaluate grading systems based on a single quality measure. Hence, our judgment was made considering all the evidence generated in this study, including several performance metrics, proportional distribution and repartition of aGVHD phenotypes in grading categories. This extensive comparison identified several clinically relevant strengths and weaknesses of each grading system, as detailed in the results. Importantly, Consensus grading clearly underestimated the importance of GI involvement, particularly in stage 4 GI aGVHD, and hence is used with decreasing frequency in the HCT community. However, it is still referred to as today's standard in the reporting of HCT data to transplant registries, which likely impairs benchmarking. Our data showed that MAGIC grading performed best among the examined conventional grading systems. Nevertheless, this study also revealed some shortcomings that remained unnoticed. For instance, MAGIC grade III assembles a diverse set of aGVHD phenotypes, which would likely profit from reclassification, as some of these are associated with

significantly better outcomes, and MAGIC is very similar to other systems, particularly the Minnesota grading system. Despite the unquestionably important role of GI involvement in aGVHD, our data indicate that MAGIC grading may overestimate its impact. Patients with GI stage 1 are categorized into MAGIC grade II, and patients with GI stage 2 and no other organ involvement are considered grade III. The impact of multiorgan involvement is structurally underestimated with this logic. PC1 grading addressed both issues by attributing a high yet not exclusive weight to GI involvement while integrating patients' multiorgan aGVHD phenotypes. Additionally, for refined grading steps (e.g., with PC1 6 or 12 grades), PC1 grading always considers multiorgan involvement at each grade. The challenge of implementing refined data-driven grading steps in clinical practice can be eased via digital tools, such as our provided online calculator (www.gvhd.online), and can yield more precise diagnoses allowing future trials to potentially adapt treatments to the severity of aGVHD. Despite the apparent complexity of the data, they exhibited a linear variation in the multi-dimensional space. In a higher dimensional dataset, i.e., with more features including biomarkers, nonlinear methods may add further value to data-driven aGVHD grading.

This study has strengths and limitations. It included an unprecedented cohort size and level of detail in the assessment of aGVHD with effective data-driven methods for aGVHD assessment, which are prepared for further adaptations to the needs of precision medicine. However, its geographical setting in Europe may have limited ethnic diversity. Due to the enrollment period, the cohorts included only 1-2% of patients receiving posttransplant cyclophosphamide (PTCy); nevertheless, data-driven methods evaluating aGVHD phenotypes are equally applicable in the PTCy setting, while the clinical outcome association remains to be confirmed. GVHD treatment data were not reported in detail. In its present form, the PC1 algorithm may slightly underestimate the gravity of isolated single organ stage 4 aGVHD involvement for the skin or liver; however, this limitation is unlikely to restrain clinicians from treating aGVHD and may be overcome with even larger datasets. Data-driven grading currently uses clinical aGVHD assessments but may also be extended by integrating biomarkers or other data sources from precision medicine.

This study provided a detailed analysis of all aGVHD grading systems and pioneered and demonstrated the utility of data-driven grading, particularly PC1 grading. These results support HCT registries in policy change to inform more comprehensively on aGVHD. HCT centers should not only report the overall aGVHD grade to registries but also provide each organ's involvement for optimal benchmarking. Clinical trials should report more details on the involvement of specific organs, heterogeneity, and distribution of aGVHD phenotypes in their cohorts, data which may be analyzed by these tools to improve HCT care. Data-driven grading provides an expandable platform for classification and risk assessment supporting precision medicine in transplantation.

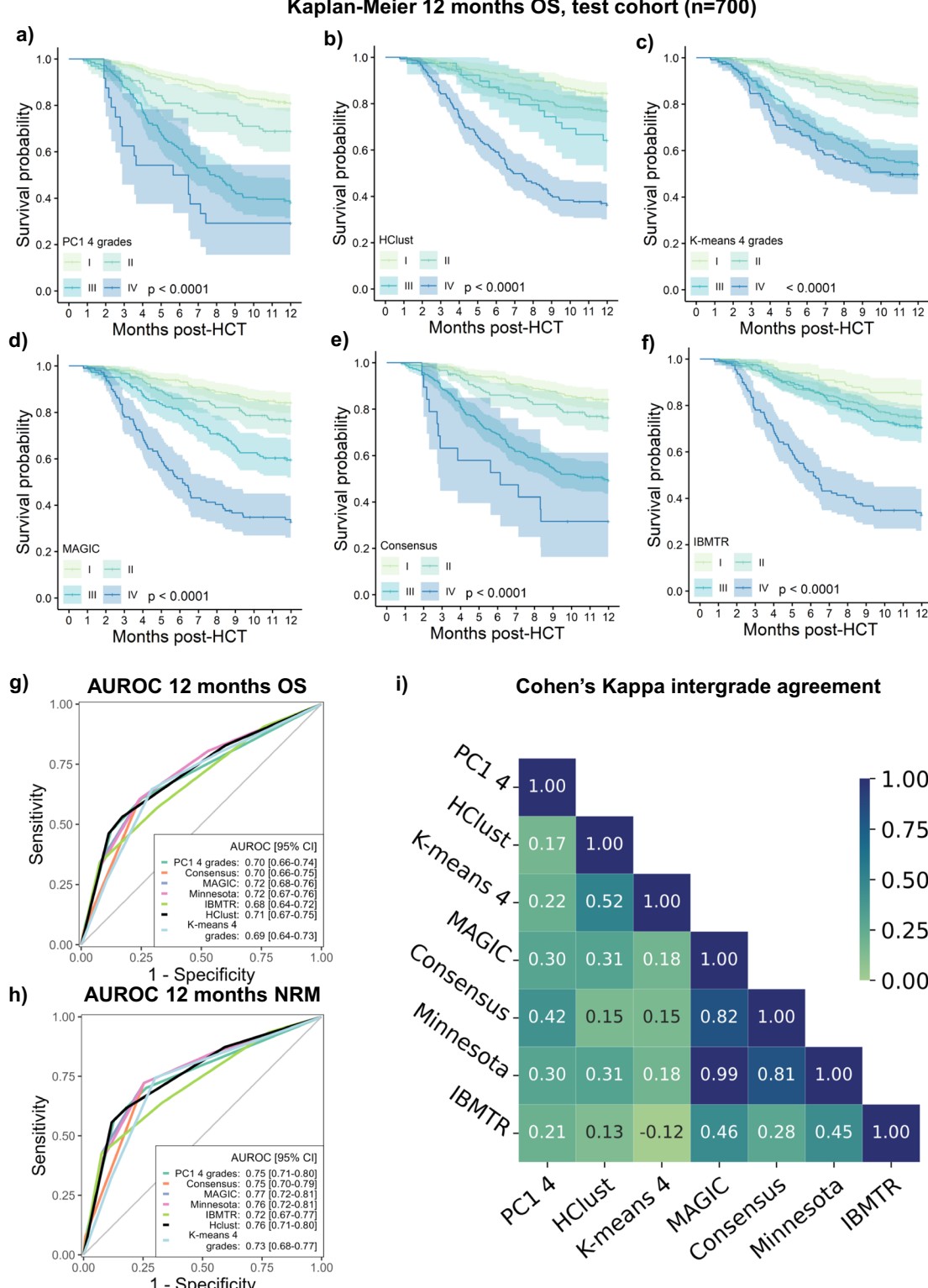

**Fig. 5 | Comparative visualization of data-driven and conventional grading methods on the independent test cohorts (*n* = 700).** Comparison of OS between aGVHD classifications using four grades (both conventional and data-driven grading), Cohen's analysis of intergrading agreement and assessment of the respective predictive values using AUROC. **a–f**: Kaplan–Meier OS curves with 95% CI of patients in the independent test cohorts (*n* = 700). OS is stratified by aGVHD grading severity according to the relevant grading system from I to IV and strata are compared with the two-sided log-rank test. **a** PC1-aGVHD grading, **b** Hierarchical clustering grading (Hclust) **c** K-means clustering grading with 4 grades **d** MAGIC grading, **e** Consensus grading, **f** IBMTR grading. **g** Comparison of AUROC for 12 months OS between grading systems. AUROC values range from 0.5 to 1.0. **h** Comparison of AUROC curves for 12 months NRM. **I**: Cohen's Kappa comparing the intergrading agreement of different grading systems (PC1, Hclust, *K*-means, MAGIC, Consensus, IBMTR, and Minnesota). Ranges from 0 (no agreement, light green) to 1 (full agreement, dark blue).

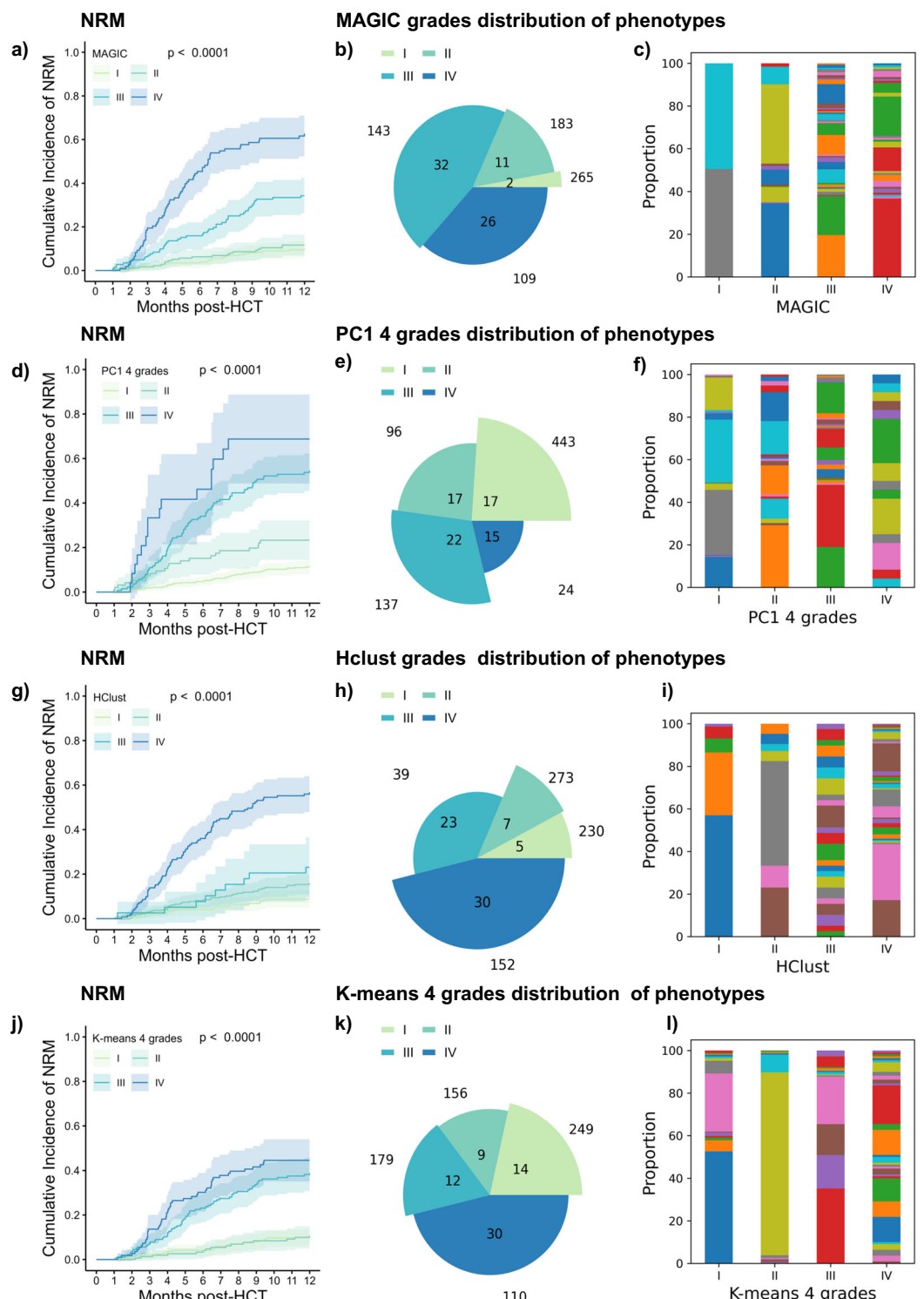

**Fig. 6 | Comparative distribution analysis of aGVHD grading methods on the independent test cohort (*n* = 700).** Data-driven aGVHD grading methods are compared to MAGIC conventional grading to reveal differences in patient proportions, organ combinations involved in each grade and ability to dissect into cohorts with significantly distinct NRM. **a** Cumulative incidence NRM curves according to MAGIC grades. Separation of curves is tested by the two-sided Gray test. Error bands represent 95% CI **b** Pie chart of aGVHD grades in MAGIC grading. The angle of each slice is proportional to the number of organ stage combinations in the respective grade. The radius of each slice represents the number of patients within this grade. MAGIC grade I: 2 combinations and 265 patients, II: 11 combinations and 183 patients, III: 32 combinations and 143 patients, IV: 26 combinations and 109 patients. **c** Patient phenotype distribution within each grade. Stacked bar chart of MAGIC aGVHD grades showing the proportion of patients in each organ

stage combination. For each bar the color represents one combination, no cross-over between grades. All phenotypes are detailed in Supplementary Data 1. **d** Cumulative incidence NRM curves according to PC1 grading with 4 grades. Separation of curves is tested by two-sided Gray test. Error bands represent 95% CI **e** Pie chart of PC1 grades. **f** Stacked bar chart of PC1 grades. The phenotypes are detailed in Supplementary Data 2. **g** NRM of Hclust grading with four grades. **h** Pie chart of Hclust grades. **i** Stacked bar chart of Hclust grades. The phenotypes are detailed in Supplementary Data 3. **j** Cumulative incidence NRM curves according to K-means grading using 4 grades. Separation of curves is tested by a two-sided Gray test. Error bands represent 95% CI **k** Pie chart of *K*-means grades. **l** Stacked bar chart of *K*-means grades. The phenotypes are detailed in Supplementary Data 4. Source data for **b**, **c**, **e**, **f**, **h**, **i**, **k** and **l** are provided as a source data file.

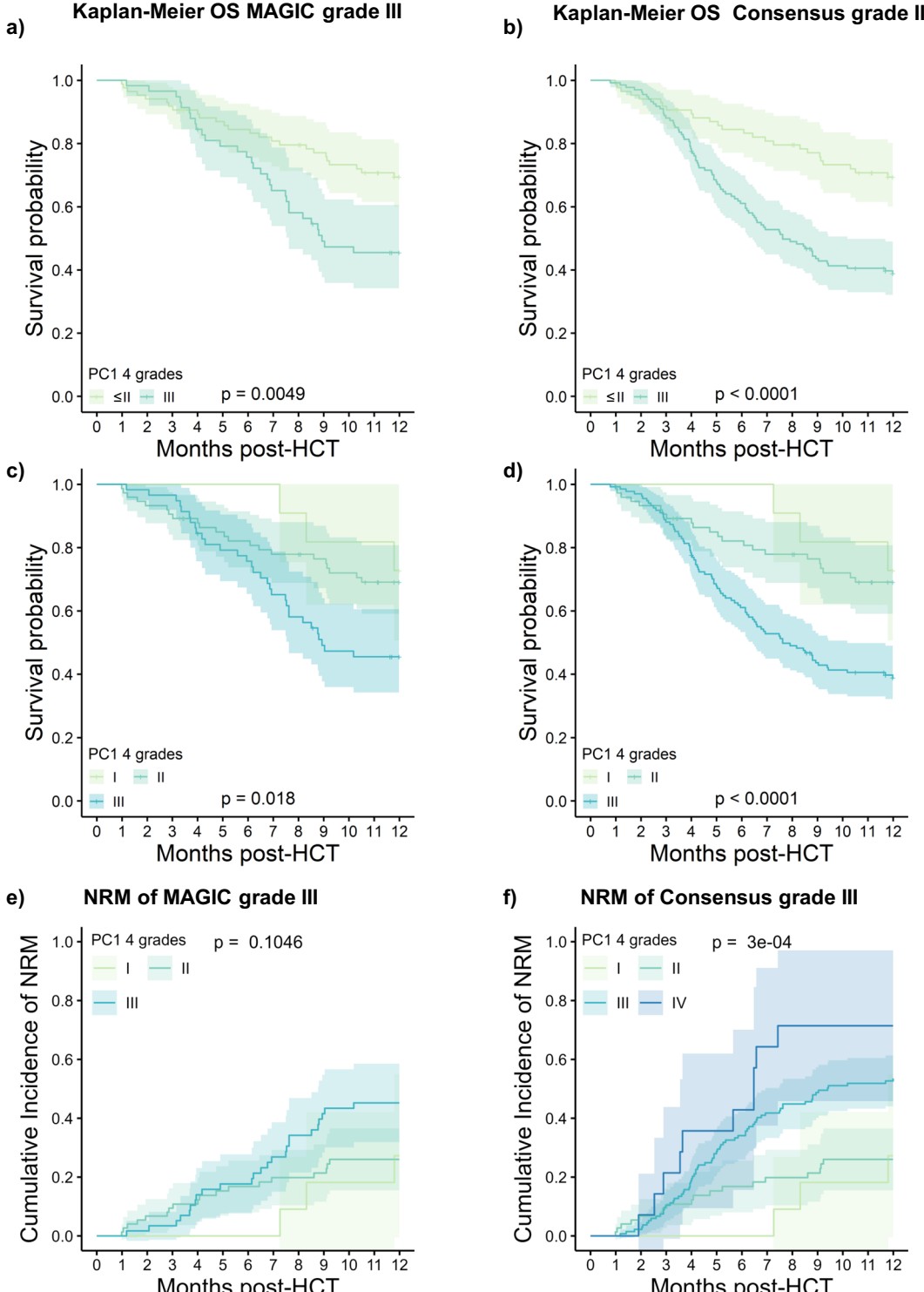

**Fig. 7 | Clinical outcome analysis of re-distributed patients between data-driven and conventional grading systems.** Redistributed patients from one severity category to another between different grading systems are compared to the remaining patients in the original category. **a** Kaplan–Meier OS curves with 95% CI of redistributed patients from MAGIC grade III to PC1-grade ≤II (light green) compared to intersection grade III patients in both grades (dark green). Strata are compared with the two-sided log-rank test. The phenotypes are detailed in Supplementary Data 8. **b** Comparison of Kaplan-Meier OS curves with 95% CI of redistributed patients from Consensus grade III to PC1-grade ≤ II (light green) to intersection of grade III patients in both consensus and PC1 (dark-green). Strata are

compared with the two-sided log-rank test. **c** Kaplan–Meier OS curves with 95% CI of redistributed patients from MAGIC grade III to PC1-grade I (light green) and PC1-grade II (green) are compared to grade III patients in both MAGIC and PC1 (blue-green). **d** Kaplan–Meier OS curves with 95% CI of redistributed patients from Consensus grade III to PC1-grade I (light green), to PC1-grade II (green) are compared to the intersection of grade III patients (blue-green). **e** Cumulative incidence curves of NRM are compared for the same strata as in c. Error bands show 95% CI. **f** Cumulative incidence curves of NRM are compared for redistributed Consensus grade III patients to PC1 including PC1-grade IV (dark blue). Error bands show 95% CI. Strata for NRM are compared with the two-sided Gray test.

**Fig. 8 | Outcome analysis according to aGVHD target organ severity in the test cohort ($n = 700$).** Patients in the test cohort ($n = 700$) were stratified according to target organ severity staging. **a** and **b** Kaplan–Meier OS and cumulative incidence of NRM of patients stratified by aGVHD skin stage 0–4. Error bands represent 95% CI. Separation of curves is tested by the two-sided log-rank test (OS) or two-sided Gray test (NRM). **c-d** Kaplan–Meier OS and cumulative incidence of NRM of patients stratified by aGVHD liver stage 0–4. Error bands represent 95% CI. Separation of curves is tested by the two-sided log-rank test (OS) or two-sided Gray test (NRM). **e** and **f** Kaplan–Meier OS and cumulative incidence of NRM of patients stratified by aGVHD GI stage 0–4. Error bands show 95% CI. Separation of curves is tested by the two-sided log-rank test (OS) or two-sided Gray test (NRM).

## Methods

### Patients

This analysis included 3754 adult patients with allogeneic HCT between January 2008 and December 2018, of which 3019 had aGVHD. The training cohort of data-driven aGVHD grading included 2319 consecutive patients diagnosed with aGVHD after HCT from two major German academic HCT centers, the Department of Hematology and Stem Cell Transplantation of the West German Cancer Center at University Hospital Essen ($n = 1345$) and the Department for Stem Cell Transplantation of the University Medical Center Hamburg-Eppendorf

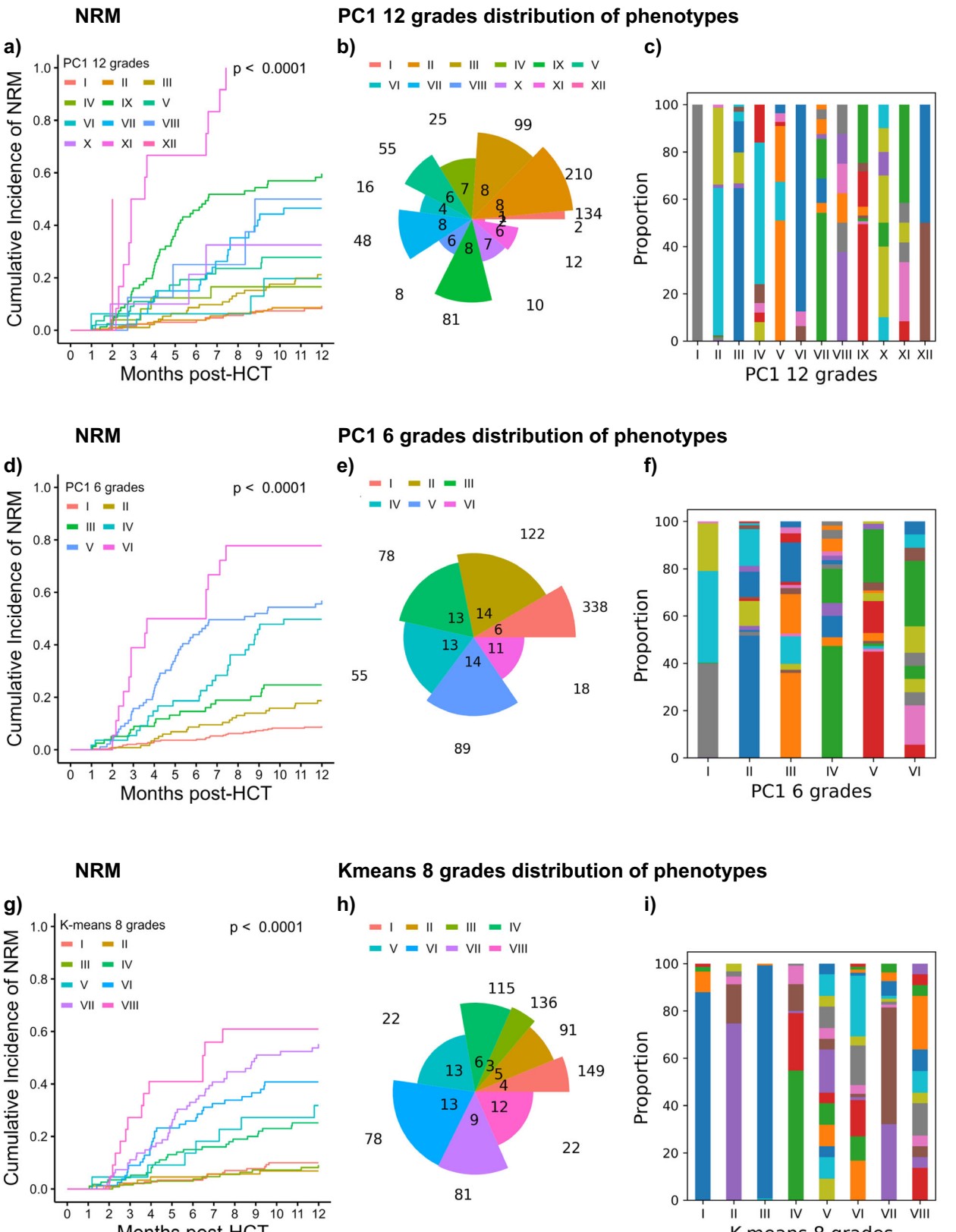

(n = 974). The independent test cohort included 700 patients from 3 large academic HCT centers, the Department of Hematology, Hemostasis, Oncology and Stem Cell Transplantation of Hannover Medical School (n = 434), the Department of Medicine and Hematology, Charité Universitätsmedizin Berlin (n = 156) and the Department of Internal Medicine V, University Hospital Heidelberg (n = 110). For

comparative Cox analysis of different aGVHD grading systems, an additional cohort (n = 735) of HCT patients without aGVHD (aGVHD grade 0) served as the reference group. Patient baseline data, donors, HCT characteristics, aGVHD organ stages and HCT outcomes were extracted from patients' electronic health records and retrospectively analyzed. To ensure data quality, correct classification and

**Fig. 9 | Comparative distribution analysis of aGVHD grading systems refined beyond four grades on the independent test cohort (n = 700).** Additional data-driven aGVHD gradings with more than 4 grades are compared to reveal differences in patient proportions, organ combinations involved in each grade and their ability to dissect into cohorts with significantly distinct NRM. **a** Cumulative incidence NRM curves of aGVHD grades in PC1 grading using all 12 PC1 stages as distinct severity grades. Colors representing PC1 aGVHD grade I–XII. Separation of curves is tested by the two-sided Gray test. **b** Pie chart according to PC1 with 12 grades. The angle of each slice represents the number of organ stage combinations in the respective grade. The radius of each slice represents the number of patients within this grade. **c** Stacked bar chart of PC1 with 12 grades showing the proportion of patients in each organ stage combination. For each bar, one color represents one combination, no crossover between grades. The phenotypes are detailed in Supplementary Data 9. **d** Cumulative incidence NRM curves according to PC1 grading with 6 grades. Colors representing PC1 aGVHD grade I–VI. Separation of curves is tested by the two-sided Gray test. **e** Pie chart of PC1 with six grades. **f** Stacked bar chart of PC1 with six grades. The phenotypes are detailed in Supplementary Data 10. **g** Cumulative incidence NRM curves according to K-means grading with eight grades, using the optimal number of clusters as determined by elbow method on the development cohort. Colors representing K-means aGVHD grade I–VIII Separation of curves is tested by the two-sided Gray test. **h** Pie chart of K-means-8 grades **i** Stacked bar chart of K-means-8 grades. The phenotypes are detailed in Supplementary Data 11. Source data for **b**, **c**, **e**, **f**, **h** and **i** are provided as a source data file.

standardized aGVHD reporting between different HCT centers, full documentation of clinical aGVHD organ involvement (skin, liver, GI) was required and served to calculate the grades of both conventional and data-driven classifications. Appropriate coverage of aGVHD phenotypes in cohorts was verified during the exploratory data analysis. Assignment to HCT treatment, GVHD prophylaxis, supportive care and follow-up care were based on standardized clinical treatment protocols and HCT center policies, considering hematologic diagnosis, age, comorbidities and donor constellation. Details are provided in the supplementary data. After HCT, inpatients were screened daily for clinical signs of aGVHD. Outpatients were assessed at each visit (i.e., weekly/biweekly during the first 2 months). Further outpatient follow-up intervals were sequentially extended, depending on clinical status and HCT-related complications. Once diagnosed, aGVHD severity was evaluated daily for inpatients and at each visit for outpatients and documented in patients' health records. The maximum extent of aGVHD organ involvement and severity was used as an input variable for analysis.

## Data-driven analyses of aGVHD phenotypes, development of grading and internal validation

Exploratory data analysis was performed on the training (n = 2319) and test cohorts (n = 700) to ensure adequate data distribution in the independent test cohort. A correlation matrix was generated with Spearman's rank correlation coefficients for the target organ stages (skin, liver, and GI). Pair plot analysis was used to illustrate and compare the distribution of single variables of the skin, liver, and GI and their mutual relationship.

Principal component analysis (PCA) was applied to transform the multidimensional dataset into a set of successive orthogonal components so that the variance in the data could be explained at best in a lower dimensional space. Biplot and scree plots were used to display the results of this dimensionality reduction. The loading scores of the first principal component (PC1, principal component with the largest eigenvalue) were used to formulate an aGVHD severity index $s_i'$

$$s_i' := (\vec{\mathbf{p}}_i - \vec{\mathbf{p}}_0) \cdot \vec{\mathbf{c}}_1 + s_0 \qquad (1)$$

where $\vec{\mathbf{p}}_i \in V$ represents each patient's aGVHD stage, $\vec{\mathbf{p}}_0$ is the centering term, $\vec{\mathbf{c}}_1$ refers to the loadings of PC1, and $s_0$ is an offset due to centering in the PCA algorithm. PC1 aGVHD stage $s_i$ is defined as follows:

$$s_i = f(s_i') := \mathrm{round}(s_i' * 2) + s_0' \qquad (2)$$

where $s_i'$ is the severity index, and $s_0'$ is an offset to shift $s_i$ to start from the minimum stage 1 to a maximum stage of 12. To compare data-driven with conventional grading, the number of possible aGVHD grades was set to 4. Thus, the PC1 aGVHD stages $s_i$ were linearly grouped into 4 PC1 grades with PC1 stages 1-3 as PC grade I, 4–6 as grade II, 7–9 as grade III and 10–12 as grade IV. During PCA, the results were internally validated via 500-fold bootstrapping, with each sample consisting of 1546 randomly selected data points (=2/3 of the training cohort). The mean, minimum and maximum of each eigenvalue from bootstrapping were used as validation metrics (Supplementary Fig. 2c).

As alternative data-driven methods, agglomerative hierarchical clustering (HClust) and partitional K-means clustering were applied to the training cohort data. For HClust, a bottom-up approach was adopted, and the distance threshold was set to 30 to split the cohort into 4 clusters. Distances between data points for the clusters were determined by ward linkage using the minimum increase of the sum of squares (MISSQ) as previously described[27]. The results are presented in a dendrogram. Partitional K-means clustering was the second alternative data-driven method that we applied. The determination of the optimal cluster combination was evaluated via both the sum of squares of distances (SSD, elbow method) and the silhouette coefficient performance index. Again, the default cluster number was set to 4. Cluster numbers >4 were used for supplemental analysis (e.g., K-means 8 grades). Given that clustering-defined grades were not automatically categorized into higher or lower severity, we ranked clustering grades using OS. Despite its large size, the training cohort did not represent all possible aGVHD phenotypes (i.e., organ involvement constellation). aGVHD phenotypes of the test cohort that were not defined by the training cohort (n = 6) were omitted for validation.

## External test and verification of data-driven aGVHD grading systems

All studied aGVHD grading systems in this study were externally tested on an independent multicenter test cohort from three large German HCT centers. To compare the grading performance of different data-driven and conventional aGVHD grading systems, we employed several established comparative metrics.

We applied the Akaike information criterion (AIC) as previously described[28] using the following formula:

$$\mathrm{AIC} = -2 \times \log(L) + 2 \times p' \qquad (3)$$

Where $\log(L)$ is the conditional log-likelihood for Cox proportional hazards models and $p'$ is the number of categories in a given grading system. Due to the potential presence of tied events, the Efron method[29] was additionally used. The range of the AIC varies with the size of the studied population but is comparable in the same dataset. Smaller AIC values are preferable.

Harrel's concordance index (Ci)[30,31] was employed to compare the prognostic ability of the survival models. The risk groups in the survival models were created with respect to the aGVHD grades, which were obtained by using data-driven and conventional grading methods. The range of the Ci is from 0 to 1, with a higher value considered superior.

Time-dependent receiver operating characteristic (ROC) curves were generated to visualize the specificity and sensitivity of aGVHD classifications for 12-month OS and NRM. The areas under the ROC curves (AUROCs) were compared between different grading methods.

a)

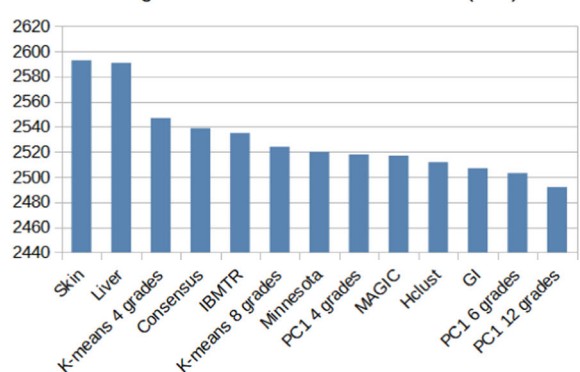

b)

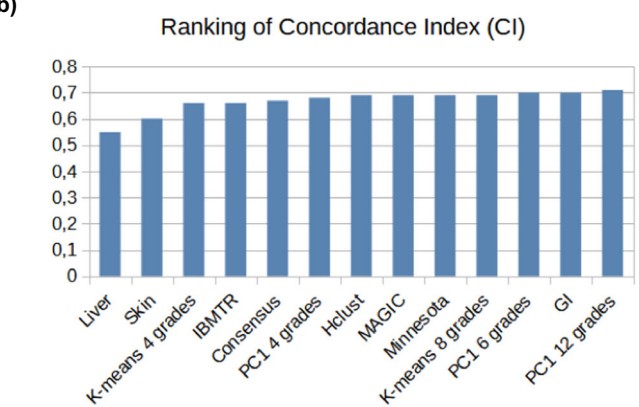

c)

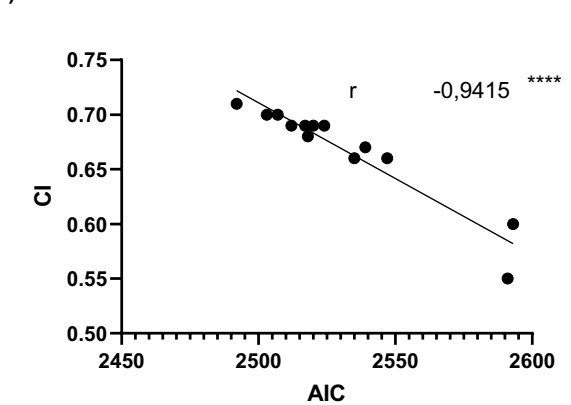

**Fig. 10 | Comparison of classification performances using Akaike information criterion and concordance index. a** Bar plot visualization of the Akaike information criterion (AIC) of all aGVHD grading combinations in decreasing order. If not otherwise mentioned, patients are categorized into four grades. Lower AIC results are preferable. **b** Bar plot visualization of the concordance index (c-index/CI) of all aGVHD grading combinations in increasing order. Higher c-index values are preferable. **c** AIC plotted versus CI for all analyzed aGVHD classification methods. Correlation (r) calculated via linear regression. 95% CI: −0.98 to 0.81 ****$p < 0.0001$ (two-tailed). As the clustering-based grading systems did not cover the phenotype constellation of $n = 6$ patients in the test cohort, the comparison between systems was performed among the remaining 694 patients. Source data are provided as a source data file.

Cohen's kappa was calculated as previously described[32] to compare the intergrading agreement of the different grading systems, which categorize patients into 4 severity grades (PC1, HClust, *K*-Means, MAGIC, Consensus, Minnesota and IBMTR). Cohen's kappa ranges from 0 (no agreement) to 1 (full agreement). Values between 0.41 and 0.6 indicate moderate agreement, and values between 0.61 and 0.8 indicate substantial agreement. Values between 0.81 and 1 indicate very high agreement.

The external test cohort ($n = 700$) of aGVHD patients was analyzed for nonrelapse mortality (NRM) and overall survival (OS) at 12 months. Patient outcomes were comparatively visualized for each grading system with stratification according to the respective aGVHD severity grade. OS was analyzed via Kaplan–Meier analysis[33]; aGVHD grade subgroups were compared using the log-rank test; and survival hazards were calculated by a Cox proportional hazards model[34]. NRM and relapse were considered competing events and were analyzed by competing risk analysis and compared by Gray's test. *P* values < 0.05 were considered statistically significant.

### Implementation
The development of data science grading methods (Spearman, pair plot, PC1 grading, clustering methods, Cohen's kappa) was performed with the open-source software Python (version 3.10.2) using the following libraries: scikit-learn, numpy, scipy, pandas, matplotlib, seaborn, os, pyaml and datetime. Nonlinear dimensional reduction methods tSNE and UMAP, clinical outcome analysis and AIC calculations were performed with R[35] (version 4.1.3, R Core team, Vienna, Austria, 2020, https://www.r-project.org) using the following libraries: Rtsne, umap, tidyverse, readxl, writexl, ggplot2, survival, survminer, cmprsk, ggstatsplot, cmprsk, dplyr, dynpred, tidyr, aod, tableone and timeROC. Details and references to all libraries are provided in the supplementary data.

### Ethics
Study protocol approval was obtained by the institutional review board of the University Duisburg-Essen (Protocols Nos. 17-7675-BO and 21-9965-BO). All patients have given written informed consent to the collection, electronic storage, and scientific analysis of anonymized HCT-specific patient data in accordance with German legislation and the revised Helsinki Declaration. We confirm that no patient can be identified through the use of anonymized patient data.

### Reporting summary
Further information on research design is available in the Nature Portfolio Reporting Summary linked to this article.

## Data availability
An online calculator for comparative calculation of data-driven grading systems is provided free of charge under www.gvhd.online. To enable independent replication of our methods, we included detailed descriptions of data-driven grading systems development in the "Methods" section. Processed source data for individual figures are provided in this paper. Anonymized datasets generated during the current study are available upon request. Requests can be addressed to the corresponding author (amin.turki@uk-essen.de; expected response time 2 weeks). The individual clinical raw data contains sensitive personal health information, are protected and are not available due to data privacy laws. Collective anonymized clinical data are available under restricted access due to sensitive personal health information, access will be provided via the University Hospital Essen and are subject to approval by the data protection officer and ethics committee and formalized via data access agreements. Source data are provided with this paper.

## Code availability

Source code for PC1-derived grading has been deposited in (https://github.com/AGchi/data-driven-grading-of-aGVHD) accompanied by instructions.

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

## Acknowledgements

This study was supported by the Bundesministerium für Bildung und Forschung (BMBF) through grant 031L0027 (DWB) by the Deutsche Forschungsgemeinschaft (DFG) grant FU 356/12-1 (ATT), by the Stiftung Universitätsmedizin Essen (ATT) and by the open access publication fund of the University Duisburg-Essen via project DEAL. OP acknowledges the support of DFG (PE 1450/7-1, PE 1450/9-1, 1450/10-1). GC acknowledges AIRC Foundation (Associazione Italiana per la Ricerca contro il Cancro, Milan, Italy) grant 26216 and the IMI 2 HARMONY/HARMONY+ projects 116026. Dr. Lambros Kordelas, Dr. Marcel Wiesweg, and Dr. Nils Leimkühler corrected drafts of the manuscript. Dr. Pietro Crivello is thanked for provided critical advice.

## Author contributions

A.T.T. designed and supervised the study. A.T.T., D.W.B., F.A.A., and N.K. performed data collection for the training cohort and provided clinical expertise. G.B., O.P., T.L., N.B., and N.K. provided test cohort data and clinical expertise. T.G. and E.B. performed data analysis, data-driven aGVHD grading model development and statistical analysis. E.B., T.G., and A.T.T. interpreted the data. F.A.A. and G.C. contributed to data interpretation. A.T.T. supervised model development and wrote the manuscript. E.B., T.G., and D.W.B. contributed to writing the manuscript. H.C.R. contributed clinical expertise. All authors had access to primary data, read and approved the final manuscript.

## Funding

## Competing interests

The authors of this manuscript have potential competing interests to disclose. A.T.T. Consultancy for CSL Behring, Maat Pharma, Biomarin and Onkowissen. E.B. is an employee of Bayer AG at the time of publication of this manuscript, research was conducted before his engagement at Bayer and without the involvement of Bayer. G.B. reported no conflicts directly related to this work. G.B. collaborates with Jazz Pharmaceuticals, Shionogi, and Medac in clinical trials. O.P. has received honoraria or travel support from Gilead, Jazz, MSD, Novartis, Pfizer, and Therakos. He has received research support from Incyte and Priothera. He is a member of advisory boards to Equillium Bio, Jazz, Gilead, Novartis, M.S.D., Omeros, Priothera, Sanofi, Shionogi, and SOBI. D.W.B. received travel subsidies from Medac, all outside the submitted work. H.C.R. received consulting and lecture fees from Abbvie, AstraZeneca, Vertex, Novartis, and Merck. H.C.R. received research funding from Gilead Pharmaceuticals and AstraZeneca. H.C.R. is a co-founder and shareholder of CDL Therapeutics GmbH. O.P. has no conflicts directly related to this work. The other authors report no competing interests.

## Inclusion and ethics statement

All local contributors to this research were included as authors or acknowledged, when not fulfilling authorship criteria. We confirm that this research has local, regional, and global impact and has been designed and conducted in collaboration with local communities.
