## [Peer Review File · Nature Communications]

Data-driven grading of acute graft-versus-host diseaseREVIEWER COMMENTS

Reviewer #1 (Remarks to the Author):

This is a novel study challenging the limitation of the prediction and early intervention of acute GVHD.

#1 Training cohort vs. validation cohort

This study divided the two cohorts according to the institutions. The review considers the actual HSCT practice (protocol) in each institution and moreover the patient background can be different. Background comparison between the two cohorts should be more precisely performed.

#2 aGVHD diagnosis and treatment

The diagnosis of GVHD is shown to be done "centrally", but more detailed information or protocol should be described. Is the diagnosis is really uniformed?

Moreover, the cohort ranged from 2008 to 2018. Significant changes in treatments occurred during this period. Can such treatment options be adjusted in this model?

Those without aGVHD can offer also important data on survival in comparison to those with significant aGVHD.

#3 haplo HSCT

The authors indicated haploidentical HSCT are included. More information is necessary to judge whether this cohort can be analyzed simultaneously.

Reviewer #2 (Remarks to the Author):

Bayraktar, Graf, et al developed data-driven grading systems for acute GVHD following allogeneic hematopoietic cell transplantation using machine learning techniques. They used large training (n = 2319) and test (n = 700) datasets to develop their algorithms. They then compared their data-driven grading systems to conventional grading systems (Consensus, MAGIC, etc.) and found that acute GVHD

grading using algorithms developed with machine learning techniques may improve stratification of outcomes over conventional grading systems. This stratification may offer advantages regarding prognosis and treatment decisions. The manuscript is dense in terms of computational results and iteration between models but is well written and proposes a welcome advance to acute GVHD grading. To my knowledge the methodology appears sound and the online calculator makes these new algorithms easily accessible. It is excellent to see further use of machine learning techniques to help with data interpretation and clinical decision-making in allogeneic hematopoietic cell transplantation.

Reviewer #3 (Remarks to the Author):

Baryaktaer and colleagues present a commendable study in which they utilize a remarkable multicenter cohort to develop and validate data-driven approaches for acute GVHD grading/classification systems, while comparing them to existing ones. The authors employ multiple unsupervised learning techniques to aggregate possible GVHD organ system involvement in the development cohort. The resulting models are then assessed for associations with mortality across cohorts and compared to established systems. The data-driven approach demonstrates an improved classification of GVHD states compared to the traditional 4-grade classification schemes. Additionally, a stronger relationship between the new GVHD states and the clinical outcomes of the patients is observed.

The paper is well-written and comprehensive, with a large sample size of 3019. The approach used for training and validation is reasonable, wherein the validation set consists of centers not included in the development set.

However, there are several critiques that should be addressed:

1. While the paper presents multiple computational approaches to define GVHD grades, it lacks a clear message regarding which model clinicians should use. The readers may become lost in the details.
2. The features of the cohorts, including baseline patient, disease, donor, and transplant characteristics, should be presented. Additionally, outcomes in each cohort, such as GVHD and survival, need to be described.
3. The authors evaluate multiple unsupervised learning techniques for GVHD classification, including PCA, hierarchical clustering, K-means, UMAP (with DBSCAN variation), and tSNE. However, these non-linear methods may confuse readers and could either be moved to the supplementary materials or omitted altogether, as they do not significantly contribute to the paper.
4. The different classification schemes are correlated with survival days relative to the day of transplantation. However, GVHD is an event that occurs after transplantation and should therefore be considered as a time-dependent covariate. Moreover, solely looking at survival days is inappropriate as it ignores censoring. Any presentation of survival in days should be supported by KM estimates. Associations between GVHD grading and survival/NRM should be backed up by regression modeling,

considering GVHD as a time-dependent covariate. Finally, associations between GVHD grading and outcomes should be presented using multivariable adjustment within regression models to strengthen the analysis.

5. AUC estimates should be provided with 95% confidence intervals.

6. The appropriateness of using AUCROC for time-to-event outcomes should be addressed.

7. The section on "Verification and test on independent data" is lengthy and could be made more concise.

8. In Figure 4C, it is stated that the optimal number of clusters determined by both methods is 8. However, a higher Silhouette value and lower SSD are observed with a higher number of clusters. Please clarify this inconsistency.

9. The website provided is inactive and cannot be evaluated.

Point-by-point response to peer review:

Reviewer #1 (Remarks to the Author):

This is a novel study challenging the limitation of the prediction and early intervention of acute GVHD.

Response: Thank you for your positive evaluation of our study and for your suggestions to further improve this manuscript.

#1 Training cohort vs. validation cohort

This study divided the two cohorts according to the institutions. The review considers the actual HSCT practice (protocol) in each institution and moreover the patient background can be different. Background comparison between the two cohorts should be more precisely performed.

Response: As suggested, we created a patient baseline table, comparing the background characteristics of the training and validation cohorts (Suppl. Table 1). Most variables were reasonably balanced between the independent cohorts. The detected differences between the training and validation cohorts, e.g. higher proportion of AML in the validation cohort, however, did not negatively impact the overall applicability of the PC1-derived grading and underline its generalizability. We also included these variables in the multivariate Cox regression models (Figure 4e and Suppl. Figure 6) and added a description of practice standards of the included centers and on each institution's particularities to the supplementary methods.

#2 aGVHD diagnosis and treatment

The diagnosis of GVHD is shown to be done "centrally", but more detailed information or protocol should be described. Is the diagnosis is really uniformed?

Response: Indeed, we classified each patient "centrally" with an R code categorizing the documented organ involvements according to the respective aGVHD grading systems to avoid potential bias related to inconsistencies in classification practice. The clinical diagnosis of GVHD, however, was established by the treating physicians at each contributing institution and thus has not been centrally uniformed. It has to be emphasized, however, that the participating institutions have long-lasting and comprehensive experience in the diagnosis and clinical grading of aGVHD using commonly accepted diagnosis criteria of aGVHD.

Moreover, the cohort ranged from 2008 to 2018. Significant changes in treatments occurred during this period. Can such treatment options be adjusted in this model? Those without aGVHD can offer also important data on survival in comparison to those with significant aGVHD.

Response: Following the recommendation of reviewer 1, we created additional multivariate models for NRM (Figure 4E) and OS (Suppl. Figure 6), adjusted for the treatment year as continuous variable, which indeed significantly associated with outcome. In these multivariate models adjusted for 11 covariates including treatment year, the main effect of the PC1-derived data-driven aGVHD grades remained significant at high hazards. The impact of the treatment year on outcome is in line with published reports (e.g. Gooley et al. NEJM 2010, Penack et al. Blood Adv. 2020, Greinix et al. Haematologica 2022) and we recognize that overall therapeutic, supportive and donor selection HSCT options improved in particular from the 1990s and 2000s. However, previous reports also showed that in recent periods this improvement was less important (Greinix et al. Haematologica 2022). We deliberately chose the inclusion period between 2008 and 2018, because of its relative stability in many relevant HSCT practices. At the beginning of the inclusion period,

azole fungal prophylaxis was readily established in most European centers represented in this study as well as high resolution HLA typing. Furthermore, the included centers used similar protocols for antiviral and aGVHD prophylaxis during this period e.g. T cell depletion with ATG in patients with higher GVHD risk (Detailed in supplementary methods). The second-line trial and approval of Ruxolitinib for the treatment of steroid-resistant GVHD occurred after 2018. Also, the practice of haploidentical HSCT with PTCy increased after 2018 in many European centers. All patients received high-dose steroids as first-line aGVHD treatment. Unfortunately, the detailed further-line aGVHD treatments have not been collected for all patients of this cohort, yet in principle, the model could be adjusted for treatment options in cohorts with documented treatment. We agree with reviewer 1 on the role of patients without aGVHD as comparators and therefore used a cohort of patients without aGVHD as common reference in some of our Cox regression models to compare different grading algorithms (previous Suppl. Figure 5, revision Suppl. Figure 8).

#3 haplo HSCT

The authors indicated haploidentical HSCT are included. More information is necessary to judge whether this cohort can be analyzed simultaneously.

Response: Given the inclusion period and the setting in Germany, only about 1-2% of patients of our study received a haploidentical HSCT, this information is now included in the baseline table (Suppl. Table 1). To rule out bias from the remaining haplo patients, we did an additional multivariate subgroup analysis of the test cohort excluding haploidentical HCT patients (n=690) using the PC1 grading, which again figured as significant covariate at a high hazard ratio (e.g. PC1 grade III HR 5.62, 95%CI 3.92-8.07; data not shown in the manuscript). In principle the data-driven grading algorithms should be equally applicable to all types of transplant settings as they focus on classifying aGVHD severity, which in conventional grading practice is also handled similarly for MUD and haplo HSCT. With increasingly available data from haplo HSCT with PTCy it will be possible in future studies to confirm the association of our data-driven grading with clinical outcome in this setting. Due to the increasing relevance of haplo HSCT with post-transplant cyclophosphamide prophylaxis we stated the limitation that “the cohorts included only few patients receiving post-transplant cyclophosphamide (PTCy), nevertheless these data-driven methods evaluating aGVHD phenotypes are equally applicable in the PTCy setting, while the clinical outcome association remains to be confirmed.” on page 15 line 427.

Reviewer #2 (Remarks to the Author):

Bayraktar, Graf, et al developed data-driven grading systems for acute GVHD following allogeneic hematopoietic cell transplantation using machine learning techniques. They used large training (n = 2319) and test (n = 700) datasets to develop their algorithms. They then compared their data-driven grading systems to conventional grading systems (Consensus, MAGIC, etc.) and found that acute GVHD grading using algorithms developed with machine learning techniques may improve stratification of outcomes over conventional grading systems. This stratification may offer advantages regarding prognosis and treatment decisions. The manuscript is dense in terms of computational results and iteration between models but is well written and proposes a welcome advance to acute GVHD grading. To my knowledge the methodology appears sound and the online calculator makes these new algorithms easily accessible. It is excellent to see further use of machine learning techniques to help with data interpretation and clinical decision-making in allogeneic hematopoietic cell transplantation.

Response: Thanks for your positive evaluation of our study and efforts to help advance the field.

Reviewer #3 (Remarks to the Author):

Bayraktar and colleagues present a commendable study in which they utilize a remarkable multicenter cohort to develop and validate data-driven approaches for acute GVHD grading/classification systems, while comparing them to existing ones. The authors employ multiple unsupervised learning techniques to aggregate possible GVHD organ system involvement in the development cohort. The resulting models are then assessed for associations with mortality across cohorts and compared to established systems. The data-driven approach demonstrates an improved classification of GVHD states compared to the traditional 4-grade classification schemes. Additionally, a stronger relationship between the new GVHD states and the clinical outcomes of the patients is observed.

The paper is well-written and comprehensive, with a large sample size of 3019. The approach used for training and validation is reasonable, wherein the validation set consists of centers not included in the development set. However, there are several critiques that should be addressed:

Response: Thank you for your positive evaluation of our work and for your recommendations to further improve this manuscript.

1. While the paper presents multiple computational approaches to define GVHD grades, it lacks a clear message regarding which model clinicians should use. The readers may become lost in the details.

Response: The number of details likely resulted from this study's pioneering role in the development and validation of data-driven approaches to the grading of aGVHD. While we think that the PC1-derived grading system offers many advantages, we also acknowledged alternative methods as well as conventional grading. Following the recommendation of reviewer 3, we have adjusted the abstract, shortened and changed the results for greater clarity and added several multivariate regression models. Among the data-driven approaches to the grading of aGVHD, we recommend the PC1-derived grading system, which allows to cover 12 different severity stages (and yielded to the both the highest AIC and CI in the test cohort) to complement clinicians in their routine aGVHD assessments. Thus, the online calculator is focused on the PC1 derived grading.

2. The features of the cohorts, including baseline patient, disease, donor, and transplant characteristics, should be presented. Additionally, outcomes in each cohort, such as GVHD and survival, need to be described.

Response: As suggested, we have created an additional baseline table of relevant features (Suppl. Table 1). Most variables were reasonably balanced between the independent cohorts. The detectable differences between training and validation cohorts, e.g. higher proportion of AML in validation cohort, however, did not negatively impact the applicability of the PC1-derived grading and underline its generalizability. These features were also integrated as covariates into multivariate regression models (Figure 4e, Suppl. Figure 6). Regarding the outcomes, we previously plotted the Kaplan Meier OS curves both of the training (Figures 1-4d) and of the validation cohort (Figures 5-9) in the main manuscript. We now also added Supplementary Table 2 comparing clinical outcome parameters between training and test cohort.

3. The authors evaluate multiple unsupervised learning techniques for GVHD classification, including PCA, hierarchical clustering, K-means, UMAP (with DBSCAN variation), and tSNE. However, these

non-linear methods may confuse readers and could either be moved to the supplementary materials or omitted altogether, as they do not significantly contribute to the paper.

Response: Following the suggestions of reviewer 3, we have shortened and adjusted this section to avoid any confusion. The related methods as well as the experimental results using non-linear methods were moved to the supplementary methods and supplementary results (including Suppl. Figures 3-5).

4. The different classification schemes are correlated with survival days relative to the day of transplantation. However, GVHD is an event that occurs after transplantation and should therefore be considered as a time-dependent covariate. Moreover, solely looking at survival days is inappropriate as it ignores censoring. Any presentation of survival in days should be supported by KM estimates. Associations between GVHD grading and survival/NRM should be backed up by regression modeling, considering GVHD as a time-dependent covariate. Finally, associations between GVHD grading and outcomes should be presented using multivariable adjustment within regression models to strengthen the analysis.

Response:

We have taken up these recommendations and added to this revised manuscript multivariate regression models using the PC1-derived aGVHD grades as time-dependent covariates for NRM (Figure 4e) and OS (Suppl. Figure 6). These multivariate models were adjusted for other potentially confounding variables, which were stepwise integrated as covariates and the main effect of the PC1-derived data-driven aGVHD grades remained significant at high hazards. In addition, we provided comparative NRM cumulative incidence curves from the date of aGVHD diagnosis in Supplementary Figure 7. In Figure 3 c-d, which reviewer 3 referred to, we previously plotted the PC1 axis against survival in days, which however was primarily intended for illustrative purpose to spread the data points and as a proxy for long-term OS, not as feature for model development. All previous “proper” OS analyses were performed using KM estimates with confidence intervals (e.g. Figure 3f). For greater clarity, we adjusted the legends of Figures 3c-d and also state that “censoring has not been considered in this representation”. We also agree that a mere outcome association (e.g. with survival) as a means to evaluate classifications has many limitations, why we applied several other metrics such as e.g. the Akaike information criterion within our evaluation of the grading systems.

5. AUC estimates should be provided with 95% confidence intervals.

Response: Done. Figure 5 g-h has been updated with these results.

6. The appropriateness of using AUCROC for time-to-event outcomes should be addressed.

Response: We agree with reviewer 3 that AUROC can be potentially problematic for time-to-event outcomes and that these should be used with caution. We therefore had a) chosen the timeROC library for analysis, which was specifically built for time-to-event data and b) critically addressed this issue in our manuscript, results section “However, given the examined features of aGVHD phenotypes, a simple clinical outcome association alone does not answer the question of optimal grading” on p. 9 line 242. Following the suggestion of reviewer 3 we added “Despite careful consideration of the time-dependent character of aGVHD in these models, the clinical outcome

association alone may not be sufficient to evaluate classifications” to the results section on p.9 line 256.

7. The section on "Verification and test on independent data" is lengthy and could be made more concise.

Response: We have revised and shortened the respective section, as recommended.

8. In Figure 4C, it is stated that the optimal number of clusters determined by both methods is 8. However, a higher Silhouette value and lower SSD are observed with a higher number of clusters. Please clarify this inconsistency.

Response: We agree that the graph suggests another cutoff at 14 clusters. We, however, had determined the optimal cluster number of 8 using the elbow method as no significant improvement of SSD was observed from this point. Since our algorithm was designed to identify the first significant clustering by these methods, we pursued further analyses using 8 clusters.

Yet, for identifying the optimal number of clusters, there is no one-size-fits-all solution, reason why we followed the suggestion of reviewer 3 and additionally explored the "second" K means cutoff with 14 clusters. It yielded to a competitively low AIC of 2516 and a high CI of 0.71 on the test cohort, yet did not discriminate the hazards of clusters in a linearly ascending manner with increasing severity. Furthermore, many small clusters had comparable outcomes (Suppl. Figure 9), hence we did not see sufficient added value to retain it as core result, also to avoid confusing the reader with additional details. We described the results from this additional K means clustering with 14 clusters in the supplementary results section on supplementary page 5 and added to the legend of Figure 4 “We evaluated a further cutoff point with 14 clusters in the supplementary results.”

9. The website provided is inactive and cannot be evaluated.

Response: Apologies, this was a misunderstanding. We intentionally hosted the demo calculator during the peer review process under the provisional domain <http://tblankenheim.de/> (which is fully accessible during peer review) to avoid pre-publication.

REVIEWERS' COMMENTS

Reviewer #1 (Remarks to the Author):

Properly revised.

No further comments.

Reviewer #2 (Remarks to the Author):

Appreciate the author's comments to reviewer questions and the associated manuscript revisions. As with my initial review, I have no questions or concerns about the manuscript.

Reviewer #3 (Remarks to the Author):

All critiques have been properly addressed.

R2-Point-by-point response to peer review:

Reviewer #1 (Remarks to the Author):

Properly revised.
No further comments.

Reviewer #2 (Remarks to the Author):

Appreciate the author's comments to reviewer questions and the associated manuscript revisions. As with my initial review, I have no questions or concerns about the manuscript.

Reviewer #3 (Remarks to the Author):

All critiques have been properly addressed.

Response: We thank all the reviewers for their positive evaluation of our study as well as for their constructive feedback and suggestions, which we think have further improved the present manuscript.

Following the recommendation from the Nature Communications editorial office this manuscript has been additionally reviewed by a native speaker editing service and finalized according to the editorial policy, resulting in the additional tracked changes of the main text.